# Exploring the impact of college graduates' place attachment on entrepreneurial intention upon returning to hometowns: A study based on the theory of planned behavior

**Cixian Lv[1], Jingjing Xu[2]\*, Wenhao Chang[3], Xiaotong Zhi[1], Peijin Yang[1], Xinghua Wang[1,4]\***

**1** Normal College, Qingdao University, Qingdao, China, **2** Institute of Education, Xiamen University, Xiamen, China, **3** Physical Science and Technology College, Ningbo University, Ningbo, China, **4** College of Business Management, Xiamen Huaxia University, Xiamen, China

\* xujj9703@163.com (JX); xinghua_wang689@qdu.edu.cn (XW)

**Data Availability Statement:** Study participants were asked for their consent to sharing their de-identified data; this process and the consent form

## Abstract

The issue of the continuing decline of rural areas caused by urbanization has become a global concern. Encouraging college graduates to return to their hometowns to start businesses is an important initiative for countries to achieve sustainable rural development. Drawing from the Theory of Planned Behavior (TPB), this study introduces two additional variables: place attachment and entrepreneurial self-efficacy. Through a series of three model refinements, a comprehensive theoretical framework has been formulated to elucidate Chinese college graduates' hometown-based entrepreneurial intention and behavior. The samples for this study were 1151 college graduates selected from diverse universities across China. This study aims to explore the influence of college graduates' hometown-based entrepreneurial intention using Structural Equation Modelling. This analytical approach illuminates how variables such as college graduates' place attachment, entrepreneurial self-efficacy, subjective norm for hometown-based entrepreneurship, and attitude towards hometown-based entrepreneurship affected their hometown-based entrepreneurial intention. The research findings reveal the following insights: (1) The overall levels of college graduates' place attachment and hometown-based entrepreneurial intention were relatively low. (2) College graduates' place attachment, entrepreneurial self-efficacy, subjective norm for hometown-based entrepreneurship, and attitude towards hometown-based entrepreneurship, had a positive impact on their hometown-based entrepreneurial intention. (3) College graduates' place attachment and subjective norm for hometown-based entrepreneurship had a significant impact on their hometown-based entrepreneurial intention through the mediating variable of entrepreneurial self-efficacy. This study then makes policy recommendations from theoretical and managerial aspects.

was approved by the Ethics Committee of Qingdao University Data. A de-identified dataset, which can be used to replicate the study findings, may be obtained from qykeyanke@163.com.

**Funding:** The research leading to these results has received funding from the National Social Science Foundation (Education) Project (http://www.nopss.gov.cn/), grant number BIA190164 (awarded to Cixian Lv). The funders had no role in study design, data collection and analysis, decision to publish, or preparation of the manuscript.

**Competing interests:** The authors have declared that no competing interests exist.

# 1. Introduction

The issue of the continuing decline of rural areas caused by urbanization has become a global concern [1, 2]. Entrepreneurial activity is recognized as a vital endeavor that merits encouragement due to its significant contributions to the economic and social development of a given area. It serves as a means of creating employment opportunities and plays an essential role in addressing this issue [3]. For example, China has been steadfast in its pursuit of the 'rural revitalization' strategy since 2017. The primary objective of this strategy is to rectify the growing imbalance between rural and urban areas, ultimately striving for more sustainable development. A pivotal approach is attracting the population and other factors to rural regions [4]. Consequently, promoting the return of college graduates to their hometowns to initiate businesses is a crucial initiative for nations seeking sustainable rural development.

Due to two decades of rapid economic growth following the initiation of economic reforms in 1978, China has undergone significant urbanization, driven by the largest rural-to-urban migration in history [5, 6]. Statistically, the proportion of the nation's population dwelling in cities has increased from 17.9% in 1978 to 58.5% in 2017 [7]. This trend has sparked an increasing demand for greater attention to the challenges faced by rural China [8]. For example, in March 2014, the government issued the National New-type Urbanization Plan (2014–2020), which advocates that urbanization should be changed from being land-centered to people-oriented [9, 10].

The promotion of entrepreneurial behavior stands as a significant matter of shared concern among governments and scholars across the globe [11]. The academic community has conducted a substantial number of studies related to the interpretation of entrepreneurial behavior. Among these, the Theory of Planned Behavior (TPB) holds a prominent position [12, 13]. There is a consensus in the literature regarding the validity of intentional models for predicting the entrepreneurial drive, as intentions are understood as antecedents of actual behavior [14]. To enhance the college graduates' entrepreneurial behaviors after returning to hometown, the paramount objective revolves around identifying the variables that exert influence—directly or indirectly—over the entrepreneurial intentions of these graduates.

Hometown-based entrepreneurship or returning to hometown for (RHF) entrepreneurship is the behavior of a specific group of people to start businesses in specific areas, which includes two behavioral decisions: returning to their hometowns and starting businesses. Entrepreneurship usually refers to the establishment of new businesses. Rural return is a form of population migration. Studies have shown that China's household registration system and special hometown attachment have a significant impact on population migration [15]. After analyzing entrepreneurship research related to population migration, we have found that existing studies mainly focus on rural returnees' entrepreneurship [16], migrants' entrepreneurship in the migration areas, or entrepreneurial behavior that occurs in a specific region [17]. The studies on hometown-based entrepreneurship usually adopt the traditional "survival-economy" analysis paradigm but have not interpreted people's emotional motivations for returning to their hometowns [18].

The questionnaires for this study were distributed to 1363 college graduates, who were selected using the convenient sampling method from nine different types of colleges and universities. These nine universities are situated in the western, central, and eastern regions of China, which further enhances the credibility of the research results. These students encompass graduates from different academic backgrounds, such as junior college, undergraduate, and graduate studies. Given that employment choices hold greater significance for these graduates, they are likely to contemplate whether to return to their hometowns for entrepreneurial endeavors.

The contribution of this study is as follows. Returning to Hometown for (RHF) entrepreneurship is the focus of our study. The ongoing decline of rural areas caused by urbanization was the source of this problem. While numerous domestic and foreign investigations have concentrated on college students' entrepreneurial intentions and behaviors, there have been fewer studies centered around 'Returning to Hometown For' (RHF) entrepreneurship. Furthermore, the influencing mechanism of subjective norm, attitude, and entrepreneurial self-efficacy to intention for RHF entrepreneurship has been verified by a considerable number of studies. In addition to considering that aspect, this study incorporates the Theory of Man-Land Relationship into TPB. From the perspective of environmental psychology, this study has modified the TPB model by incorporating two factors including place attachment and self-efficacy, and employed the TPB model in interpreting college graduates' hometown-based entrepreneurial intention.

This study has employed the AMOS software to verify the fitting of the structural equation model and the research hypotheses. This study aimed to answer three questions: (1) What factors affect college graduates' hometown-based entrepreneurial intention? (2) What is the mechanism by which these factors affect college graduates' hometown-based entrepreneurial intention? (3) How to effectively intervene in college graduates' hometown-based entrepreneurial intention and behavior?

## 2. Theoretical foundations and research hypotheses

### 2.1 The theory of planned behavior and RHF entrepreneurship

As the mainstream theory in explaining and predicting behavior, TPB can explain and predict entrepreneurial intention and behavior to some extent [19]. According to TPB, the occurrence of individuals' whole behaviors first requires the emergence of behavioral intention, and behavior arises from the combined effects of behavioral intention and perceived behavioral control. However, the emergence of behavioral intention is based on the individual's comprehensive rational balance of internal and external factors. Among others, the most important internal and external factors include subjective norm, attitude, and perceived behavioral control. Subjective norm represents explicit norm and belief such as social pressure felt by individuals. Attitude refers to an individual's judgment of the pros and cons of behavior. Perceived behavior control is a reflection of the level of "non-motivation factors" such as the availability of necessary resources, indicating human control over behavior [12].

Entrepreneurial intention is usually considered the most reliable predictor of actual entrepreneurial behavior. The precondition for entrepreneurial behavior to occur is the emergence of behavioral intention. Therefore, in this study, the "intention" and "behavior" in the TPB model are combined and simplified into the entrepreneurial intention.

As for the relationship between subjective norm and intention for RHF entrepreneurship, the impact of subjective norm on behavioral intention has been supported by TPB, and the Technology Acceptance Model, etc. However, when it comes to the specific field of entrepreneurship, the conclusions of existing studies are not consistent. Most studies support that subjective norm affects entrepreneurial intention [20], while some studies argue that the impact of subjective norm on entrepreneurial intention is not significant [19]. One explanation is that there are cultural differences concerning the impact of subjective norm on individual behavioral intention. Influenced by Confucianism and socialism, Chinese culture emphasizes collectivity and norms, so the role of subjective norm may be more typical. Based on the above analysis, a hypothesis is proposed as follows.

H1: Subjective norm for RHF entrepreneurship has a significant positive impact on intention for RHF entrepreneurship.

About the relationship between attitude and entrepreneurial intention, some studies have demonstrated that attitude significantly affects entrepreneurial intention [21]. This conclusion is consistent with that of a study taking Austrian students as samples. In addition, another study using Russian students as samples has also demonstrated that attitude significantly affects entrepreneurial intention [22]. Based on the above analysis, a hypothesis is proposed as follows.

H2: Attitude toward RHF entrepreneurship has a significant positive impact on intention for RHF entrepreneurship.

As for the relationships among perceived behavioral control, self-efficacy, and intention for RHF entrepreneurship, perceived behavioral control in TPB reflects the availability of the opportunities and resources necessary for an individual to conduct a certain behavior. It describes the level of objective conditions for the behavior to occur. However, in reality, entrepreneurial opportunities and resources are not fixed; their availability is usually closely related to an individual's perception, thinking, and judgment. Moreover, the higher availability of opportunities and resources is not simply equal to the better conditions under which the behavior is likely to occur. On the contrary, self-efficacy refers to an individual's confidence or belief in his/her ability to achieve behavioral goals in a specific field, and it can more comprehensively and accurately reflect the individual's overall perception of the level of external conditions [23]. Some studies have compared the components of self-efficacy and perceived behavioral control. They noted that self-efficacy may be an essential predictor of behavior and the strongest predictor of behavioral intention [24]. Based on this, the TPB model was modified for the first time by using self-efficacy to replace the variable of perceived behavior control in the TPB model. A hypothesis is proposed as follows.

H3: Entrepreneurial self-efficacy has a significant positive impact on intention for RHF entrepreneurship.

## 2.2 Place attachment and RHF entrepreneurship

"Rural return" is a form of population migration. The concept similar to rural returnees in the West is known as return migration. Different from the West, rural return in the Chinese context specifically refers to the behavior that individuals who hold a rural household registration return to the rural areas where they once left. RHF entrepreneurship refers to the rural returnees who choose to start a business in their hometown rather than elsewhere. The essence of this is that when deciding to start a business, the rural returnees have a preference for a specific location. The Push-Pull Theory suggests that individuals migrate in order to improve their lives, and that the areas of migration are influenced by the forces pushing them away from their place of origin and pulling them towards their destination [25]. Kalir's new economics of migration theory further notes that an individual's place of destination or place of origin is also affected by household conditions [26].

Place attachment refers to a positive emotional bond between a person and a specific place. This bond is expressed by the person's positive emotions of being close to the place and his/her painful feeling of being away from the place. This is a hot topic of research in the fields of psychology, geography, sociology, etc., and was first proposed by Riger and Lavrakas (1981) [27]. If place attachment is used to specifically describe an individual's attachment to his/her native land, it is also called native land bonding. The emergence of native land bonding is mainly influenced by factors such as length of residence, cultural background, education, and owning one's home [28, 29]. There is no theoretical consensus on the composition of native

land bonding. The most representative one is the five-dimension theory, in which Hammitt argues that place bonding includes place familiarity, belongingness, identity, dependence, and rootedness [30].

Place attachment affects population migration behavior [31]. Some studies have shown that there is a negative correlation between place attachment and mobility, that is, the stronger the place attachment, the lower the tendency of people to choose to migrate [32]. The longer a person stays in a place, and the more connections he/she makes, the less likely he/she is to migrate [33]. Some studies further note that if the employment problems of return migration cannot be solved, they may migrate away again. Therefore, return migrants usually choose to make investments or start businesses [34]. McCormick et al. (2003) further note that rural returnees are more likely to return to their places of origin to start businesses [35]. Mayer et al. (2017) argue that return migrants are more likely to choose to start businesses in the traditional migration regions [36]. Rupasingha and Marré (2018) note that because rural areas have advantages in some aspects, some companies are migrating from urban to rural areas [37]. In addition, some studies have shown that the probability of rural returnees starting businesses is much higher than that of residents without migrating experience [38]. Furthermore, some studies argue that individuals' place attachment affects their responsible behavior, and such individuals are more willing to actively care for and improve their hometowns [39].

Based on the above analysis, we have incorporated place attachment into the TPB model, modified the model for the second time, and made the following hypothesis:

H4: Place attachment has a significant positive impact on individuals' intention for RHF entrepreneurship.

## 2.3 The mediating effect of self-efficacy

As established in empirical research, both subjective norm [20] and attitude [21] wield significant influence on the intention for RHF entrepreneurship. Building upon this foundation and our hypotheses, we recognize the pivotal impact that place attachment may exert on the intention for RHF entrepreneurship. Additionally, prior studies has underscored the crucial role of self-efficacy as a predictor of behavior and the strongest determinant of behavioral intention [24, 40]. Entrepreneurial self-efficacy, in particular, has been identified as a significant influencer of entrepreneurial intention [41].

Moreover, our review of existing literature corroborates the following points:

Entrepreneurship education can shape entrepreneurial self-efficacy through attitude [41]. Based on this finding, we propose the following hypothesis:

H5: Self-efficacy has a mediating effect on the relationship between attitude towards RHF entrepreneurship and intention for RHF entrepreneurship.

Secondly, place attachment has a significantly positive influence on entrepreneurial self-efficacy [42]. Accordingly, we posit the following hypothesis:

H6: Self-efficacy has a mediating effect on the relationship between place attachment and intention for RHF entrepreneurship.

Finally, subjective norm has a significantly positive influence on self-efficacy [43, 44]. Building upon this observation, we present the following hypothesis:

H7: Self-efficacy has a mediating effect on the relationship between subject norm for RHF entrepreneurship and intention for RHF entrepreneurship.

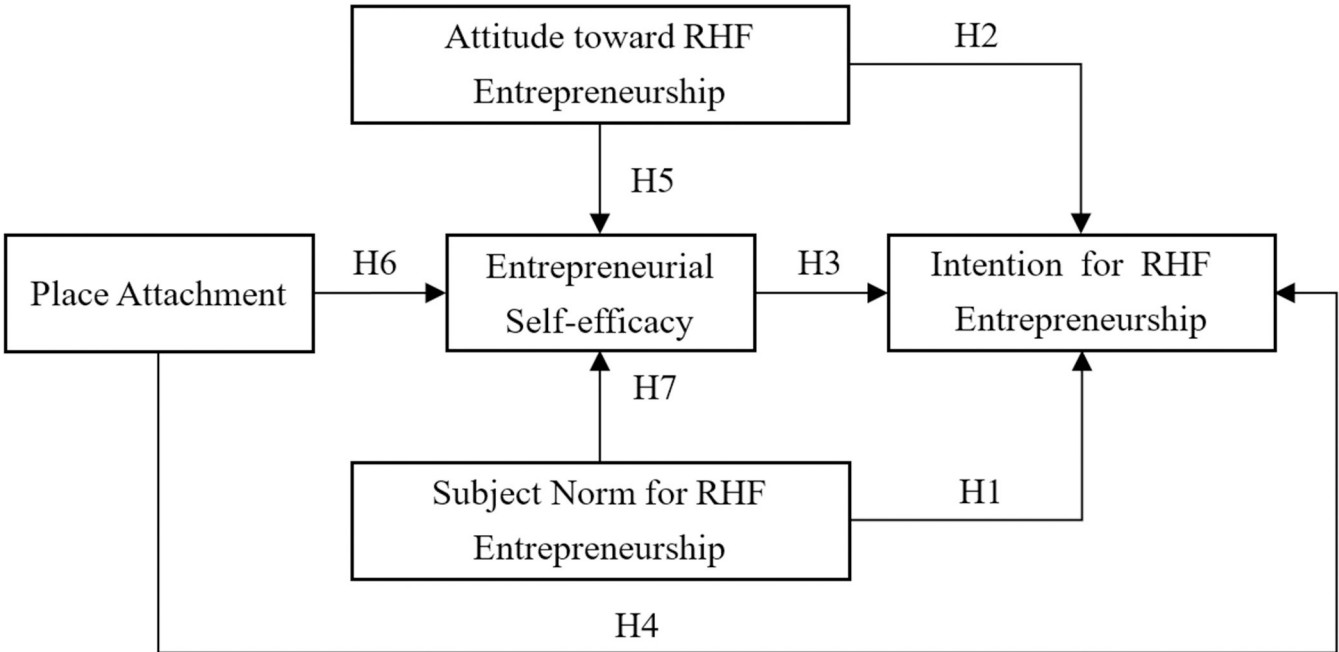

**Fig 1. Hypothesized model.**

Drawing from these analyses, we establish multiple mediating relationships with entrepreneurial self-efficacy serving as the mediating variable within the TPB model. Subsequently, we have refined the model for the third iteration, and the updated theoretical model diagram is depicted in Fig 1.

## 3. Methodology

### 3.1 Research samples and data collection

The questionnaires for this study were distributed to 1363 college graduates from October 2020 to December 2020. They were selected by employing the convenient sampling method from nine different types of colleges and universities. These nine universities are situated in the western, central, and eastern regions of China, which further enhances the credibility of the research results. These students encompass graduates from different academic backgrounds, such as junior college, undergraduate, and graduate studies. Given that employment choices hold greater significance for these graduates, they are likely to contemplate whether to return to their hometowns for entrepreneurial endeavors. The questionnaires are adapted from the mature scale. Finally, 1151 valid questionnaires were obtained, with a valid rate of 84.45%.

Among the samples surveyed, they ranged in age from 21 to 31 ($M = 24.43$, $SD = 3.81$). More specific demographics of the respondents are as follows (see Table 1).

### 3.2 Ethics statement

This project was approved by the Ethics Committee of Qingdao University. Informed written consent has been obtained. It's important to note that this research survey was entirely voluntary, which was communicated clearly to all participants. To ensure anonymity, we took measures to prevent access to any information that could potentially identify individual

**Table 1. The demographics of the respondents.**

| Variables | Level | Quantity | Ratio |
|---|---|---|---|
| Gender | Male | 696 | 60.47% |
| | Female | 455 | 39.53% |
| Academic backgrounds | Associate degree or below | 437 | 37.97% |
| | Bachelor's degree | 468 | 40.66% |
| | Master's degree or above | 246 | 21.37% |
| Professional backgrounds | Humanistic and social science | 582 | 45.94% |
| | Science and Engineering | 685 | 54.06% |
| Entrepreneurial backgrounds | Yes | 143 | 11.29% |
| | No | 1124 | 88.71% |

participants, both during and after the data collection process. Interview materials are subject to analysis only if participants have granted their consent. And we guarantee that the data is stored safely and only for the required duration. Throughout the study, only the designated researcher had access to the original data, which was subsequently disposed of upon completion of the study.

## 3.3 Measures

**3.3.1 Place attachment.** This study has developed the Place Attachment Scale based on modifying the place bonding scale [30], which mainly examines the degree of individuals' attachment to their native land. The scale includes five dimensions of place attachment: place familiarity, belongingness, identity, dependence, and rootedness. The Scale consists of 17 items, all of which are scored by the 5-point Likert scale (1 point represents "Strongly Disagree", while 5 points represent "Strongly Agree"). For example, "After leaving, my hometown is still an important place in my life", "A sense of intimacy cannot be found in the other place other than my hometown", "I am familiar with most parts of my hometown", and "I rely on my hometown". The higher the score for an individual is, the higher the degree of the individual's attachment to his/her native land is. The reliability of the scale in this study is good. The Cronbach's alpha coefficient for the scale is 0.925, and the Cronbach's alpha coefficients for the dimensions are between 0.916 and 0.934.

**3.3.2 Intention for RHF entrepreneurship.** This study has developed the RHF Entrepreneurial Intention Scale based on modifying the Entrepreneurial Intention Scale [45], which examines the categories and extent of individuals' RHF entrepreneurial intention. This scale includes two dimensions: career pursuit entrepreneurship and career choice entrepreneurship. Career pursuit entrepreneurial intention reflects the strength of an individual's pursuit of entrepreneurship as a form of life value, while career choice entrepreneurial intention mainly reflects the extent to which an individual regards entrepreneurship as an alternative to career choice and transition, rather than (or not fully) regarding entrepreneurship as an intrinsic value pursuit. The scale consists of 8 items, all of which are scored by the 5-point Likert scale (1 point represents "Strongly Disagree", while 5 points represent "Strongly Agree"). For example, "Returning to hometown for entrepreneurship is my career goal", and "If the job is not deal, I will return to hometown for entrepreneurship". The higher the score for an individual is, the higher the tendency of the individual's intention for RHF entrepreneurship is. The overall Cronbach's alpha coefficient for the scale is 0.867, and the Cronbach's alpha coefficients for career pursuit entrepreneurship and career choice entrepreneurship are 0.879 and 0.854, respectively.

**3.3.3 Subjective norm for RHF entrepreneurship.** This study has developed the Scale of Subjective Norm for RHF entrepreneurship with a reference to the subjective norm measurement recommendations proposed by Ajzen et al. (1991) [12]. The Scale consists of 3 items in a single dimension, for example, "Those who are important to me suggest I should start a business in my hometown", and "Those who are important to me hope me to start a business in my hometown". All the items are scored by the 5-point Likert scale (1 point represents "Strongly Disagree", while 5 points represent "Strongly Agree"). The higher the score for an individual is, the higher the influence of the individual's subjective norm for RHF entrepreneurship is. The Cronbach's alpha coefficient for the scale is 0.881.

**3.3.4 Attitude towards RHF entrepreneurship.** This study has developed the Scale of Attitude Toward RHF Entrepreneurship by consulting the attitude measurement recommendations proposed by Ajzen et al. (1991) [12]. The scale consists of 5 items in a single dimension, for example, "It is beneficial to start a business in my hometown", and "It is wise to start a business in my hometown". All the items are scored by the 5-point Likert scale (1 point represents "Strongly Disagree", while 5 points represent "Strongly Agree"). The higher the score for an individual, the more positive the individual's attitude towards RHF entrepreneurship. The Cronbach's alpha coefficient for the scale is 0.892.

**3.3.5 RHF entrepreneurial self-efficacy.** This study has developed the RHF Entrepreneurial Self-Efficacy Scale by considering Barbosa et al.'s (2007) entrepreneurial self-efficacy measurement recommendations [46]. The scale consists of 4 items in a single dimension, for example: "I think it is feasible to return to my hometown to start a business". All the items are scored by the 5-point Likert scale (1 point represents "Strongly Disagree", while 5 points represent "Strongly Agree"). The higher the score for an individual is, the stronger the individual's RHF entrepreneurial self-efficacy is. The Cronbach's alpha coefficient for the scale is 0.931.

## 3.4 Analyses

**3.4.1 Common method bias test.** An exploratory test was performed using Harman's single-factor test. Among others, the value of the Kaiser-Meyer-Olkin Test was 0.913. In addition, the result of Bartlett's Test of Sphericity showed that its significance P value was 0.000. The factor was rotated using the maximum variance method to obtain 14 factors with a latent root of greater than 1, and the amount of variation explained by the first factor was only 24.66%, far from reaching a critical value of 40%. Therefore, there is no serious common method bias in this study.

**3.4.2 Confirmatory factor analysis.** Before validating the structural model, confirmatory factor analysis was performed on each measurement model to test the goodness-of-fit of the measurement model. Since the result of the Chi-squared test in case of a large sample size is usually significant, and this study is a study with a large sample size, the value of the Chi-squared test is not taken into account as the goodness-of-fit index (GFI). The indicator selected by confirmatory factor analysis is the ratio of the Chi-square to the degree of freedom ($\chi2/df$), which is used to reflect the complexity of the model, and the p-value reflects the reliability of the result, CFI (comparative fit index), GFI (goodness-of-fit index), IFI (incremental fit index), NFI (normed fit index), and RMSEA (Root Mean Square Error of Approximation). The results of confirmatory factor analysis are shown in Table 2. It can be seen from Table 2 that the main goodness-of-fit indicators of the measurement models have reached an acceptable level, indicating that the sample data and the measurement model fit well and the structural model can be fitted.

To summarize, prior to commencing formal data analysis, we utilized SPSS 26.0 software to manage the questionnaire data, perform descriptive statistics, assess reliability, conduct a

**Table 2. Measurement model fitting results.**

| Latent variables | X2/df | CFI | GFI | IFI | NFI | RMSEA |
|---|---|---|---|---|---|---|
| Fit criteria | <4.000 | >0.900 | >0.900 | >0.900 | >0.900 | <0.08 |
| PA | 2.851 | 0.921 | 0.894 | 0.937 | 0.960 | 0.05 |
| ESE | 2.970 | 0.915 | 0.906 | 0.933 | 0.901 | 0.09 |
| SHE | 2.762 | 0.915 | 0.925 | 0.913 | 0.919 | 0.074 |
| AHE | 2.684 | 0.902 | 0.914 | 0.937 | 0.925 | 0.055 |
| HEI | 2.344 | 0.912 | 0.935 | 0.917 | 0.915 | 0.058 |

Note. PA, place attachment; ESE, entrepreneurial self-efficacy; SHE, subjective norm for RHF entrepreneurship; AHE, attitude towards RHF entrepreneurship; HEI, Intention for RHF entrepreneurship; CFI, comparative fit index; GFI, goodness-of-fit index; IFI, incremental fit index; NFI, normed fit index; RMSEA, root mean square error of approximation.

common method bias test, and carry out correlation analysis. Subsequently, based on the excellent goodness-of-fit results from the measurement model, we employed AMOS 26.0 software to assess questionnaire validity, confirm the adequacy of the structural equation model, and test the research hypotheses.

# 4. Results

## 4.1 Descriptive statistics

According to the statistics, college graduates have a lower-middle level of place attachment ($M = 2.34$, $SD = 0.82$), with the highest level of place familiarity ($M = 4.06$, $SD = 0.35$), and the low levels of place belongingness ($M = 2.41$, $SD = 0.62$) and place identity ($M = 2.35$, $SD = 1.02$). The overall intention level for RHF entrepreneurship is reported to be low ($M = 3.27$, $SD = 0.86$). The level of their career choice entrepreneurial intention is low ($M = 2.41$, $SD = 0.23$), while the level of their career pursuit entrepreneurial intention is high ($M = 3.60$, $SD = 0.87$).

In terms of demographic variables, age was significantly correlated with attitude ($r = 0.46$, $p<0.01$) and entrepreneurial self-efficacy ($r = 0.69$, $p<0.001$). There was significant correlation between gender and attitude ($r = 0.52$, $p<0.05$). Educational background was significantly correlated with attitude ($r = 0.61$, $p<0.01$) and entrepreneurial self-efficacy ($r = 0.58$, $p<0.01$). Therefore, age, gender and education background were included as control variables in this study.

After conducting Pearson's product-moment correlation analysis, it was found that place attachment ($r = 0.61$, $p<0.001$), subjective norm for RHF entrepreneurship ($r = 0.64$, $p<0.01$), attitude towards RHF entrepreneurship ($r = 0.68$, $p<0.01$), and entrepreneurial self-efficacy ($r = 0.73$, $p<0.001$), are significantly positively correlated with RHF entrepreneurial intention. Additionally, place attachment ($r = 0.71$, $p<0.01$), subjective norm for RHF entrepreneurship ($r = 0.67$, $p<0.001$), and attitude towards RHF entrepreneurship ($r = 0.73$, $p<0.001$), are significantly positively correlated with entrepreneurial self-efficacy. These findings provide support for the hypotheses.

## 4.2 Testing of research hypotheses

**4.2.1 TPB, place attachment and RHF entrepreneurship.** Based on an outstanding goodness-of-fit of the measurement model, this study tested the structural equation model in AMOS26.0. The results show that with respect to the structural model, $\chi2/df = 2.782$, which is within the range of 1–3, indicating that the simplicity of the model was acceptable. The value

**Table 3. Standardized path coefficients for structural models.**

| Path | Relation Hypothesis | Standardization Coefficient | SD | P | Hypothesis testing |
|------|--------------------|----------------------------|-----|-----|-------------------|
| SHE→HEI | Yes | 0.367 | 0.180 | 0.000*** | Verified |
| AHE→HEI | Yes | 0.126 | 0.199 | 0.007** | Verified |
| ESE→HEI | Yes | 0.393 | 0.247 | 0.000*** | Verified |
| PA→HEI | Yes | 0.774 | 0.160 | 0.000*** | Verified |
| SHE→ESE | Yes | 0.658 | 0.246 | 0.008** | Verified |
| AHE→ESE | Yes | 0.617 | 0.151 | 0.061 | Unverified |
| PA→ESE | Yes | 0.606 | 0.126 | 0.000*** | Verified |

Note. PA, place attachment; ESE, entrepreneurial self-efficacy; SHE, subjective norm for RHF entrepreneurship; AHE, attitude towards RHF entrepreneurship; HEI, intention for RHF entrepreneurship; Yes = influence; SD = standard deviation

**p < 0.01

***p < 0.001.

of RMSEA, which is 0.062, is less than the critical value of 0.08, which indicates that the model has a good fit. Additionally, CFI = 0.911, GFI = 0.924, IFI = 0.907, TLI = 0.945, indicating that the sample data fit well with the hypothesis model. The path coefficient significance test has shown in Table 3, as follows.

As shown in Table 3, we found that subjective norm for RHF entrepreneurship has a significant positive impact on intention for RHF entrepreneurship ($t = 0.367$, $p<0.001$), thus supporting H1. Similarly, attitude towards RHF entrepreneurship has a significant positive impact on intention for RHF entrepreneurship ($t = 0.126$, $p<0.01$), therefore substantiating H2. Additionally, entrepreneurial self-efficacy has a significant positive impact on intention for RHF entrepreneurship ($t = 0.393$, $p<0.001$), thus supporting H3. Place attachment has a significant positive impact on RHF entrepreneurial intention ($t = 0.774$, $p<0.001$), therefore substantiating H4. In conclusion, these four factors including subjective norm, attitude, entrepreneurial self-efficacy and place attachment all significantly influenced RHF entrepreneurial intention.

**4.2.2 The mediating effect of self-efficacy.** As shown in Table 3, we found that subjective norm has a significant positive impact on entrepreneurial self-efficacy ($t = 0.658$, $p<0.01$), thus supporting H5. Additionally, place attachment has a significant positive impact on entrepreneurial self-efficacy ($t = 0.606$, $p<0.001$), therefore substantiating H7. However, the effect of attitude on entrepreneurial self-efficacy was not significant ($t = 0.617$, $p>0.1$), therefore H6 is not verified. To further test the mediating effect, the Bootstrap method is employed, the results of which are shown in Table 4.

It can be observed from Table 4 that the confidence interval for testing the mediating effect of entrepreneurial self-efficacy between place attachment and RHF entrepreneurial intention is [0.121,0.204], which excludes 0, thus confirming the presence of the mediating effect. The confidence interval for testing the mediating effect of entrepreneurial self-efficacy between

**Table 4. Mediation effect analysis.**

| Path | Effect | BootLLCI | BootULCI | z | P |
|------|--------|----------|----------|-----|-----|
| AHE→ESE→HEI | 0.242 | -0.108 | 0.192 | 4.173 | 0.071 |
| PA→ESE→HEI | 0.238 | 0.121 | 0.204 | 6.232 | 0.000 |
| SHE→ESE→HEI | 0.259 | 0.095 | 0.211 | 4.325 | 0.000 |

Note. PA, place attachment; ESE, entrepreneurial self-efficacy; SHE, subjective norm for RHF entrepreneurship; AHE, attitude towards RHF entrepreneurship; HEI, intention for RHF entrepreneurship.

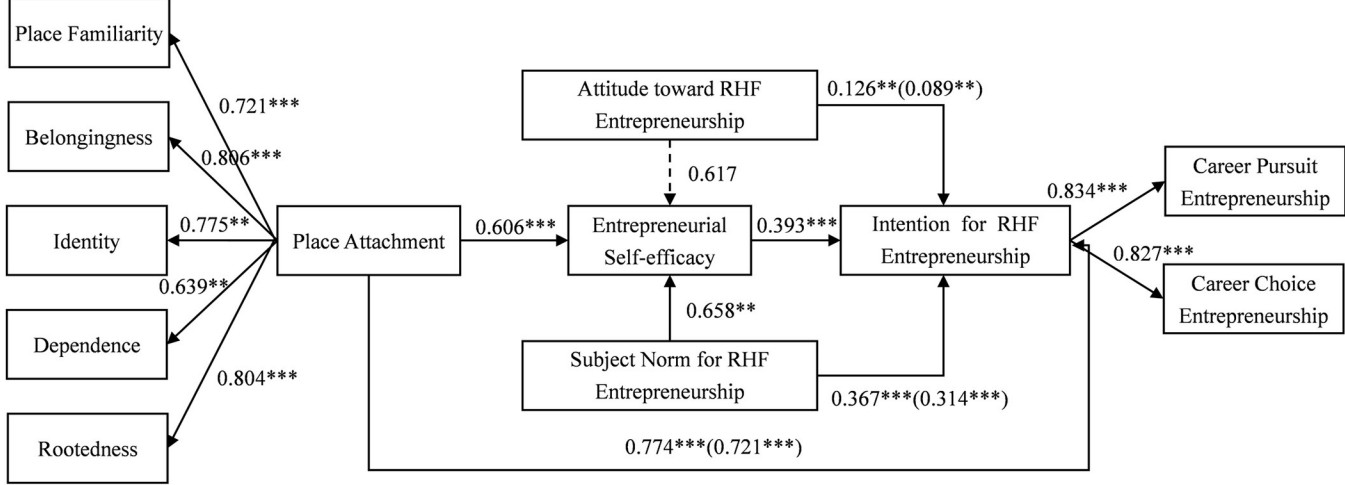

*Note.* The brackets contain standardized regression coefficients of independent variables to dependent variables after adding the mediating variables; \*\**p* < 0.01, \*\*\**p* < 0.001.

**Fig 2.  The mediating effect model.** *Note.* The brackets contain standardized regression coefficients of independence variables to dependent variables after adding the mediating variables; \*\**P*<0.01, \*\*\**P*<0.001.

subjective norm for RHF entrepreneurship and intention for RHF entrepreneurship is [0.095,0.211], which excludes 0, so the mediating effect is verified. However, the confidence interval for testing the mediating effect of entrepreneurial self-efficacy between attitude towards RHF entrepreneurship and RHF entrepreneurial intention is [-0.108,0.192], which includes 0, therefore, the mediating effect is not supported. The mediating effects of entrepreneurial self-efficacy between place attachment, attitude towards RHF entrepreneurship, or subjective norm for RHF entrepreneurship and intention for RHF entrepreneurship are illustrated in Fig 2.

To sum up, in the influence path of "attitude towards RHF entrepreneurship → entrepreneurial self-efficacy → intention for RHF entrepreneurship", attitude towards RHF entrepreneurship had no significant impact on entrepreneurial self-efficacy, thus H5 has not been verified. In the influence path of "place attachment → entrepreneurial self-efficacy → RHF entrepreneurial intention", the independent variable place attachment still has a significant impact on the dependent variable RHF entrepreneurial intention after adding the mediating variable entrepreneurial self-efficacy ($t = 0.721$, $p<0.001$). The results show that entrepreneurial self-efficacy is a partial mediator between the relation of place attachment and RHF entrepreneurial intention, thus supporting H6. Similarly, in the influence path of "subjective norm for RHF entrepreneurship → entrepreneurial self-efficacy → RHF entrepreneurial intention", the independent variable subjective norm for RHF entrepreneurship still has a significant impact on the dependent variable RHF entrepreneurial intention even with the inclusion of the mediating variable entrepreneurial self-efficacy ($t = 0.314$, $p<0.001$). The results show that entrepreneurial self-efficacy is a partial mediator between the relation of subjective norm for RHF entrepreneurship and RHF entrepreneurial intention, thus supporting H7.

## 5. Discussion and conclusion

In recent years, more than 11.2 million people in China have returned to their hometowns to start businesses or find employment [47]. Some of these groups are college graduates who have received higher education and accepted more new things. It follows that encouraging college

graduates to return to their hometowns to start businesses is an important initiative for countries to achieve sustainable rural development [48]. For example, China's Ministry of Education actively promotes rural engagement among young people through entrepreneurship competitions. College graduates' engagement in RHF entrepreneurship is affected by multiple factors [49, 50], and the complexity of the influencing factors brings about onerous challenges in the development of RHF entrepreneurship.

College graduates constitute a distinct group among those returning to their hometowns for entrepreneurship. A positive emotional bond existed between themselves and hometowns, called place attachment. Kyle notes that an individual's special feelings about a particular place can significantly influence his/her behavior, especially when that place is under threat [51]. Hidalgo and Hernandez (2001) similarly emphasize that individuals with a positive emotional bond to a specific place tend to gravitate towards it [52]. Notably, there is a negative correlation between place attachment and population migration behavior, that is, the stronger the place attachment, the lower the tendency of people to choose to migrate [32]. For college graduates, the stronger the level of place attachment, the stronger the possibility of returning to their hometowns for entrepreneurship. Similar studies in the West also supported this opinion. Immigrants are their research object, which found that returnee migrants are more likely to return to their place of origin to start business [35, 38]. Contrary to studies in the West that focus on immigrants, our research specifically addresses rural return in the Chinese context. This phenomenon pertains to individuals holding a rural household registration who return to the rural areas they once left. Based on this, we have incorporated place attachment into the TPB model, modifying the model for the second time. The results illustrated that place attachment has a significant positive impact on intention for RHF entrepreneurship. One plausible explanation is rooted in traditional Chinese farming culture, where the land is closely related to people's positive emotions. Professor Fei further contends that China is fundamentally a folk society, with geographical connections serving as reflections of blood ties. The cultural concept of 'born here and die here' fosters a fixed, predestined relationship between people and the land [53]. The stronger the attachment to one's hometown, the better they hope it will be, and RHF entrepreneurship is one of the effective means to improve it.

As the mainstream theory in explaining and predicting behavior, TPB can explain and predict entrepreneurial intention and behavior to some extent [19]. Therefore, based on TPB, this study modified the model and collected a huge amount of data to examine the mechanism of factors affecting college graduates' intention for RHF entrepreneurship. A considerable body of empirical literature, including our study, has utilized the TPB to investigate how entrepreneurial intention is determined by the personal attitude toward entrepreneurship, by how subjective norms are perceived, and also by behavioral control [54–56]. Our study also found that subjective norm and attitude has a significant positive impact on intention for RHF entrepreneurship. Different from previous studies, perceived behavior control was replaced by self-efficacy in our study, which the TPB model was modified for the first time. On the one hand, self-efficacy can more comprehensively and accurately reflect the individual's overall perception of the level of external conditions [23]. On the other hand, comparisons of self-efficacy and perceived behavioral control components in previous studies have suggested that self-efficacy may be an essential predictor of behavior and the strongest predictor of behavioral intention [24]. The results show that entrepreneurial self-efficacy has a significant positive impact on intention for RHF entrepreneurship, aligning with existing studies [57]. This result reveals that TPB has a good explanatory power in the study of Chinese college graduates' RHF entrepreneurial intention.

This study found that demographic variables, such as age and educational background, exert an influence on entrepreneurial attitude and self-efficacy. This finding aligns with a

previous study that indicated college graduates with higher ages and more extensive educational backgrounds are more adaptable [58]. The author suggests that as college graduates' age and educational background increase, so does their personal capability, leading to a more open career concept. Fueled by social and family responsibilities, individuals may increasingly seek self-realization through entrepreneurial behaviors and plans. Additionally, gender emerged as a factor influencing entrepreneurial attitudes, consistent with previous studies [59]. One plausible interpretation is that in China, individuals of different genders often assume distinct social responsibilities.

Based on the above analysis, we found that place attachment, subjective norm, self-efficacy and attitude directly affect college graduates' intention for RHF entrepreneurship. This answers the first question raised in this study, that is, "What factors affect college graduates' RHF entrepreneurial intention".

In addition, our findings highlight a positive correlation between place attachment [43] and subjective norm [44] with entrepreneurial self-efficacy. Additionally, entrepreneurship education can affect entrepreneurial self-efficacy through attitude [40]. Based on this, it shows that college graduates' place attachment and subjective norm indirectly affect their RHF entrepreneurial intention through the mediating effect of self-efficacy. Therefore, we constructed multiple mediating relationships with entrepreneurial self-efficacy as the mediating variable in the TPB model, modified the model for the third time. This answers the second question raised in this study, which is "What is the mechanism by which these factors (college graduates' place attachment, entrepreneurial self-efficacy, subjective norm for RHF entrepreneurship, and attitude towards RHF entrepreneurship) affect their intention for RHF entrepreneurship". The results show that entrepreneurial self-efficacy has a mediating effect in the two groups of influencing relationships between "place attachment → RHF entrepreneurial intention" and between "subjective norm for RHF entrepreneurship → RHF entrepreneurial intention". This conclusion is supported by existing studies [23], and it further provides empirical support for Bandura's Self-Efficacy Theory.

This study has verified that the overall level of college graduates' intention for RHF entrepreneurship ($M$ = 3.27) is relatively low. This finding aligns with Yuan et al.'s (2022) survey of 897 college graduates in Guangdong Province. This survey found that when it comes to the intention for RHF entrepreneurship, it is not high for both rural youth ($M$ = 3.12) and urban youth ($M$ = 3.12) [60]. One possible explanation for this trend is that over the past years, China's urban-rural dual structure has actually caused a huge development differentiation between urban and rural areas, and there is a large gap between the vast rural and urban areas in China in terms of resources and factors of production. The public has widely accepted and internalized the belief that cities are superior to villages, leading to cognitive stigmas about rural areas such as backwardness, ignorance, and hardship. Moreover, as for college graduates' intention for RHF entrepreneurship, their career pursuit entrepreneurial intention ($M$ = 3.60) is higher than their career choice entrepreneurial intention ($M$ = 2.41). This shows that the main forces of college graduates who return to their hometowns to start businesses are the individuals who take RHF entrepreneurship as a way of pursuing intrinsic value and achieving full potential in their lives, rather than those who see it as merely a way to earn a living or an alternative career.

In addition, the results show that the overall level of college graduates' place attachment ($M$ = 2.34) is also not high. A nuanced examination of this attachment shows that while place familiarity (M = 4.06) stands out as the highest aspect, the levels of place identity (M = 2.35) and place belongingness (M = 2.41) are comparatively lower. Morgan's developmental theory of place attachment posits that the roots of place attachment can be traced back to childhood experiences, evolving in a circular manner through the interplay of 'exploration' in the external

environment and parent-child attachment behaviors [61]. In China, hundreds of millions of farmers have moved from rural to urban areas for employment, so many rural children have inevitably become left-behind children. Consequently, many rural students in China, influenced by their early experiences, may have adopted 'leaving their rural homes' as an aspirational goal from a young age. In recent years, there has been a declining trend in the proportion of rural college students in prestigious universities in China [62], suggesting a potential shift away from the pursuit of higher education as a means to increase personal human capital among rural youth. Such upbringing may cause individuals to have a high level of place familiarity with their hometown, but a low level of place identity and belongingness.

In conclusion, the implications derived from this study underscore that effectively influencing college graduates' RHF entrepreneurial intention involves enhancing their entrepreneurial self-efficacy, raising the levels of their place attachment, subjective norm for RHF entrepreneurship, and attitude towards RHF entrepreneurship.

## 6. Implication

With respect to the third question raised in this study—"How to effectively intervene in college graduates' RHF entrepreneurial intention and behavior?", based on the above research conclusions, we make the following recommendations which can divide into theoretical and managerial categories.

### 6.1 Theoretical implications

Cultural and ideological development can be crucial manner to continuously improve the quality of the dimensions including college graduates' place attachment, subjective norm, and entrepreneurial self-efficacy. Therefore, it becomes imperative to augment college students' comprehension, identification, and confidence in local culture. This involves safeguarding and inheriting excellent local culture, as well as exploring and developing a unique culture with local characteristics. Additionally, efforts should be made to assist college students in overcoming prejudices and stigmas against rural culture, fostering psychological identity, and strengthening their connection to rural areas.

In yet another vein, colleges and universities should strengthen the education of students on ideals and beliefs. This entails aiding them in establishing a correct outlook on values and entrepreneurship, thereby cultivating an entrepreneurial mindset. Meanwhile, educational institutions should encourage college students to visit the villages, become acquainted with the villages, and discover agricultural and rural entrepreneurship opportunities.

### 6.2 Managerial implications

On the basis of the above research outcomes, we can find that more managerial actions can be used to improve college graduates' RHF entrepreneurial intention.

Primarily, efforts should focus on altering the perceived 'disadvantaged' position of RHF entrepreneurship through economic inputs, influencing college graduates' attitude towards RHF entrepreneurship. According to the "push and pull" theory of population migration, college graduates' RHF entrepreneurship is affected by the common push and pull forces of urban and rural areas. What is behind this is the location advantage resulting from the differences between cities and villages in resources and factors of production. Therefore, we should, through the investment of economic resources, strengthen the construction of infrastructure such as the internet, transportation, and logistics in rural areas, and effectively improve the hardware environment for rural entrepreneurship. We should focus on strengthening support for RHF entrepreneurship and introducing supportive credit and tax policies for RHF

entrepreneurship, so that the rural entrepreneurs who return to their hometowns to start businesses can enjoy the same or even better entrepreneurial conditions as the urban entrepreneurs. Continual improvements in the conditions for rural entrepreneurship are poised to positively influence college graduates' attitudes towards RHF entrepreneurship, consequently impacting their RHF entrepreneurial intention.

The second is to continuously improve the quality of the dimensions including college graduates' place attachment and subjective norm through campus culture development in managerial level. Specifically, there are two approaches. Firstly, we should increase the supply of cultural products and services to rural areas, strengthen the cultivation of rural humanistic spirit. By creating a repertoire of high-quality and popular cultural works that resonate with grassroots situations, a foundation for cultural enrichment is laid. In addition, we should leverage the effects of enlightenment and guidance of cultural works, and improve the level of rural cultural soft environment. We should make greater efforts to advocate policies, strengthen the guidance of public opinion, promote role model publicity, and develop the norm guidelines for encouraging college graduates to return to their hometowns to start businesses. In terms of factor allocation, fully leveraging the positive role of public opinion. This comprehensive approach aims to elevate the levels of college graduates' place attachment and subjective norm for Returning to Hometown (RHF) entrepreneurship through cultural development. In doing so, it significantly propels their RHF entrepreneurial intention and behavior, aligning with the overarching objective of sustainable rural development.

The third is to enhance college graduates' entrepreneurial self-efficacy by continuously optimizing educational practices and talent development. Colleges and universities should implement high-quality entrepreneurship education and practical experiences, improve college students' entrepreneurial intentions and capabilities, and effectively enhance their self-efficacy. Additionally, colleges and universities should create a more favorable and inclusive entrepreneurial environment, encouraging students to start businesses, and support their entrepreneurship. A positive and supportive climate for entrepreneurship can significantly enhance college graduates' entrepreneurial self-efficacy, thereby significantly promoting their RHF entrepreneurial intention and behavior.

## 7. Limitations and future research

This study is subject to the following limitations. First, this study modified the TPB model focused only on specific variables (e.g., place attachment and self-efficacy) that the research team deemed relevant to the research context. There might be additional variables influencing Returning to Hometown for Entrepreneurship (RHF) intention and behavior, with intricate relationships among various factors. Future research can continue to explore other variables that this study did not pay attention to. Second, the samples in this study originated from China, but the scales employed, such as the scales of place attachment, attitude, subjective norm, and self-efficacy, were revised based on the scales developed in Western contexts. Despite efforts in translation, reliability tests, and scale validity, there may still be limitations. Future research can consider developing the required scale instruments related to TPB following the context of mainland China.

## Author Contributions

**Conceptualization:** Cixian Lv, Jingjing Xu.

**Data curation:** Xiaotong Zhi, Peijin Yang.

**Formal analysis:** Xiaotong Zhi, Peijin Yang.

**Resources:** Cixian Lv, Jingjing Xu, Wenhao Chang.

**Supervision:** Cixian Lv, Xinghua Wang.

**Validation:** Jingjing Xu, Wenhao Chang, Xinghua Wang.

**Writing – original draft:** Xiaotong Zhi, Peijin Yang.

**Writing – review & editing:** Cixian Lv, Jingjing Xu, Wenhao Chang, Xinghua Wang.

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
