## [Decision Letter · Decision Letter 0]

24 Jul 2023

PONE-D-23-12351College graduates’ place attachment and their entrepreneurial intention after returning to hometowns – An empirical study on the improved model based on the Theory of Planned BehaviorPLOS ONE

Dear Dr. Xu,

Thank you for submitting your manuscript to PLOS ONE. After careful consideration, we feel that it has merit but does not fully meet PLOS ONE’s publication criteria as it currently stands. Therefore, we invite you to submit a revised version of the manuscript that addresses the points raised during the review process.

We look forward to receiving your revised manuscript.

Kind regards,

Tien-Chi Huang

Academic Editor

PLOS ONE

Journal Requirements:

4. Please ensure that you refer to Figure 1 in your text as, if accepted, production will need this reference to link the reader to the figure.

Additional Editor Comments:

Dear Authors,

I hope this message finds you in good health. I am writing to you with updates regarding your manuscript PONE-D-23-12351, "College graduates’ place attachment and their entrepreneurial intention after returning to hometowns – An empirical study on the improved model based on the Theory of Planned Behavior," submitted to PLOS ONE.

We have now received the required number of reviews for your manuscript. I want to inform you that based on the initial round of reviews, I have made a decision for major revisions. These revisions were based on several valuable suggestions and comments from the reviewers, which I believe will help enhance the depth and quality of your study.

Your patience and understanding during this process are greatly appreciated. I am confident that the input from our reviewers will contribute significantly to the development and refinement of your manuscript.

Reviewers' comments:

Reviewer's Responses to Questions

**Comments to the Author**

1. Is the manuscript technically sound, and do the data support the conclusions?

Reviewer #1: Partly

Reviewer #2: Yes

2. Has the statistical analysis been performed appropriately and rigorously? 

Reviewer #1: N/A

Reviewer #2: No

3. Have the authors made all data underlying the findings in their manuscript fully available?

Reviewer #1: No

Reviewer #2: No

4. Is the manuscript presented in an intelligible fashion and written in standard English?

Reviewer #1: No

Reviewer #2: No

5. Review Comments to the Author

Reviewer #1: Thank you for your suggestions regarding the research activities related to my entrepreneurial intentions. Here are the translated suggestions:

Detailed Explanation of the Overall Research Design:

a. Theoretical justification for participant selection is needed, including specific sampling methods, sample size calculations using statistical tools like G power.

b. The description of the research methodology should be presented under the "Methods" section.

c. Provide a detailed discussion on ethical considerations regarding the research participants.

d. Modify the notation of "p" in the research methodology and results to lowercase italics "p". Change "p=.000" to "p<.001".

e. Provide evidence for describing each mean value as low or high in the research results.

Restructure the Research Results:

Differentiate the research results based on hypotheses and assign them with corresponding numbers for easier readability.

Enhance the Discussion:

Instead of merely repeating the research results, provide references to relevant literature that can serve as valid evidence and support the author's thoughts.

Lastly, Include More Evidence and Data on the Issues of Urbanization in Chinese Cities:

It appears necessary to include more evidence and data regarding the problems associated with urbanization in Chinese cities.

Reviewer #2: Thank you for the opportunity to review the manuscript ‘College graduates’ place attachment and their entrepreneurial intention after returning to hometowns – An empirical study on the improved model based on the Theory of Planned Behavior’ submitted to Plos One. I enjoyed reading the manuscript and found it to be an informative study of factors that affect entrepreneurial intention.

Overall, the research addresses the relevant topic and takes as a starting point some questions that require further investigation. These research questions should be developed in a theoretical and conceptual framework. Empirical analysis has been carried out to provide evidence for the research questions, but I would highly recommend that the authors focus your review on the methodological rigour. The results are appropriately interpreted and discussed. So, the manuscript has potential, but the reasoning does not flow naturally. The paper must be improved in line with the comments below:

1. Title and abstract – The title is very descriptive but not overly wordy. It needs to go wider for a more specific representation of the study's aims.

2. Please make your abstract attractive to readers (simple sentences without any repetition) and include 2-3 sentences ready to be cited exactly as they are. In 1 paragraph, your abstract should tell the readers why the study is important (maximum 25% of the text), what you did, i.e. your methodology (maximum 25% of the text), and what you found, i.e. main research results and their major implications (50% of the text). This is very important to promote your work because of the growing trend that authors use Google search to find and cite papers based on the abstract (instead of reading the full paper).

3. Regarding the Introduction, the first two paragraphs need to be enhanced with more arguments exemplifying the importance of entrepreneurship and this research. There is a lot of published work in 2021 and 2022, such as Barba et al. (2022) https://doi.org/10.1016/j.iedeen.2021.100184 or Calderon-Milan et al. (2020), https://doi.org/10.3390/su12031079, to strengthen the narrative on this.

4. In the introduction, it is very worthwhile to introduce readers in more detail to the social and cultural context in which the university student finds himself.

5. Where are the contributions of this paper? A clear and focused reference to the innovation and uniqueness of the current study is needed in light of the large number of studies already done in this field.

6. I require that you add text for each one of your hypotheses and justify each one of them by separate and in a better manner you do now. Please try you separate the particular text that corresponds with one or another hypothesis.

7. A very important intervening variable is missing- perceived behavioral control. Please explain.

8. In the methodology, the authors did not properly present the process of using "face-to-face interviews" as a tool for gathering the information - how was it done, who were the interviewers, what was their training process, how was the interviewer's reliability tested, what were the guiding questions? That is, much more details are required than have been presented.

9. An article must be self-contained, i.e. it must contain all the information necessary for its comprehension. How the sample has been chosen and to demonstrate that it is representative of the society being studied. I would also recommend adding a table with accurate information about the technical specifications of the study and another table with the comparative demographics between the population and sample.

10. Please add the validity and reliability of the measurement instruments. It is not enough to report only the Cronbach's alpha coefficient.

11. In addition, I would recommend adding a table, that shows a list of items used to measure each of the hypotheses.

12. . In section 3 it would be better if the control variables used were explained in developing the hypothesis. Where are the results for the control variables used?

13. The use of SEM should be able to test the models that have been developed. Unfortunately, how the initial model and the final model after testing is not fully shown. It would be better if this could be done so that the placement of variables or the constructs that build them could be analyzed. Please show the variables with their dimensions.

14. The conclusion and discussion section lack significant theoretical reference to the findings. The research should relate to/correspond with theories of entrepreneurship, to the Theory of Planned Behaviour.

I hope you find the above comments useful and I wish you the best of luck with developing the paper further.

6. PLOS authors have the option to publish the peer review history of their article (what does this mean?). If published, this will include your full peer review and any attached files.

Reviewer #1: No

Reviewer #2: No

---

## [Author Response · Author response to Decision Letter 0]

18 Oct 2023

The rebuttled letter has been uploaded before which can see the detaild response to academic editor, reviewer 1, and reviewer 2.

Response to academic editor:

Point 1: Please ensure that your manuscript meets PLOS ONE's style requirements, including those for file naming.

Response 1: Thanks for your reminder. We have made the necessary changes according to the journal's style requirements.

Point 2: In your Data Availability statement, you have not specified where the minimal data set underlying the results described in your manuscript can be found.

Response 2: We are grateful for your comments and suggestions. Accatually, the data set for our research is not publicly available due to ethical concerns. However, as required, the data presented in this study is available upon request from the corresponding author. This statement has been presented in the 'Data availability statement'.

Point 3: Please include your full ethics statement in the ‘Methods’ section of your manuscript file. In your statement, please include the full name of the IRB or ethics committee who approved or waived your study, as well as whether or not you obtained informed written or verbal consent. If consent was waived for your study, please include this information in your statement as well.

Response 3: We greatly appreciate your suggestions. As requested, we have added a new section titled 'Ethics Statement' within the ‘Methods’ section. The specific revisons are as follows.

" This project was approved by the Ethics Committee of Qingdao University. Informed written consent has been obtained. It's important to note that this research survey was entirely voluntary, which was communicated clearly to all participants. To ensure anonymity, we took measures to prevent access to any information that could potentially identify individual participants, both during and after the data collection process. Interview materials are subject to analysis only if participants have granted their consent. And we guarantee that the data is stored safely and only for the required duration. Throughout the study, only the designated researcher had access to the original data, which was subsequently disposed of upon completion of the study." (See Lines 260-268)

Point 4: Please ensure that you refer to Figure 1 in your text as, if accepted, production will need this reference to link the reader to the figure.

Response 4: Thanks for your suggestion. As required, we have revised it as follows.

" The modified theoretical model diagram is illustrated in Fig 1." (See Line 238)

Response to reviewer 1:

Point 1: Theoretical justification for participant selection is needed, including specific sampling methods, sample size calculations using statistical tools like G power.

Response 1: We are grateful for the reviewer's suggestion. We have made the following revision.

" The questionnaires for this study were distributed to 1363 college graduates during October 2020 to December 2020. They were selected by employing the convenient sampling method from nine different types of colleges and universities. These nine universities are situated in the western, central, and eastern regions of China, which further enhances the credibility of the research results. These students encompass graduates from different academic backgrounds, such as junior college, undergraduate, and graduate studies. Given that employment choices hold greater significance for these graduates, they are likely to contemplate whether to return to their hometowns for entrepreneurial endeavors. The questionnaires are adapted from the mature scale. Finally, 1151 valid questionnaires were obtained, with a valid rate of 84.45%." (See Lines 93-100)

Point 2: The description of the research methodology should be presented under the "Methods" section.

Response 2: We appreciate the reviewer’s comments and suggestions. As required, we have added the description of the reseach methodlogy to the "Methods" section as follows.

“To summarize, prior to commencing formal data analysis, we utilized SPSS 26.0 software to manage the questionnaire data, perform descriptive statistics, assess reliability, conduct a common method bias test, and carry out correlation analysis. Subsequently, based on the excellent goodness-of-fit results from the measurement model, we employed AMOS 26.0 software to assess questionnaire validity, confirm the adequacy of the structural equation model, and test the research hypotheses.” (See Lines 352-357)

Point 3: Provide a detailed discussion on ethical considerations regarding the research participants.

Response 3: We are grateful for the reviewer’s comments and suggestions. The following revision has been made to the manuscript.

“This project was approved by the Ethics Committee of Qingdao University. Informed written consent has been obtained. It's important to note that this research survey was entirely voluntary, which was communicated clearly to all participants. To ensure anonymity, we took measures to prevent access to any information that could potentially identify individual participants, both during and after the data collection process. Interview materials are subject to analysis only if participants have granted their consent. And we guarantee that the data is stored safely and only for the required duration. Throughout the study, only the designated researcher had access to the original data, which was subsequently disposed of upon completion of the study.” (See Lines 260-268)

Point 4: Modify the notation of "p" in the research methodology and results to lowercase italics "p". Change "p=.000" to "p<.001".

Response 4: Thanks for your suggestion. As required, we have reviewed the relevant content of the full manuscript and revised them based on your suggestion.

Point 5: Provide evidence for describing each mean value as low or high in the research results.

Response 5: We are grateful for the reviewer’s comments and suggestions about this. As required, the mean value has been discussed in the research results, as follows.

"This study has verified that the overall level of college graduates' intention for RHF entrepreneurship (M=3.27) is not high. This finding aligns with Yuan et al.'s (2022) survey of 897 college graduates in Guangdong Province. Their study also revealed that when it comes to the intention for RHF entrepreneurship, it is not high for both rural youth (M=3.12) and urban youth (M=3.12) [58]. One possible explanation for this trend is that over the past years,…." (See Lines 527-533)

"In addition, the results show that the overall level of college graduates' place attachment (M=2.34) is also not high. Specifically, the level of college graduates' place familiarity (M=4.06) is the highest, whereas the levels of place identity (M=2.35) or place belongingness (M=2.41) are not as high. According to Morgan’s developmental theory of place attachment,…." (See Lines 545-548)

Point 6: Restructure the Research Results: Differentiate the research results based on hypotheses and assign them with corresponding numbers for easier readability.

Response 6: Thanks for your comments and suggestions. As your required, we have differentiated the research results based on the hypotheses, as follows. 

The reseach results consists of two sections. The first section is titled "Descriptive Statistics," and the second section is titled "Testing of Research Hypotheses." Within the second section, we have further divided it into two parts based on the hypotheses. Part one is titled "TPB, place attachment, and RHF entrepreneurship", which mainly reported the influence of four factors on RHF entrepreneurship. Part two is titled "The mediating effect of self-efficacy" which primarily presented the mediating effect of self-efficacy between place attachment, subjective norm, attitude and intention for RHF entrepreneurship.

Point 7: Enhance the Discussion: Instead of merely repeating the research results, provide references to relevant literature that can serve as valid evidence and support the author's thoughts.

Response 7: We really appreciate your comments and suggestions about the conclusion and discussion section. As required, we have made comprehensive adjustments to this section. For example,we have improved the readability of this section, enriched the theoretical references related to the findings, and expanded the discussion on two theories. Please refer to lines 448 to 563 for the details.

Point 8: Lastly, Include More Evidence and Data on the Issues of Urbanization in Chinese Cities: It appears necessary to include more evidence and data regarding the problems associated with urbanization in Chinese cities.

Response 8: We are grateful for the reviewer's suggestion. We have made the following revision.

" Due to two decades of rapid economic growth following the initiation of economic reforms in 1978, China has undergone significant urbanization, driven by the largest rural-to-urban migration in history [5,6]. Statisticlly, the proportion of the nation's population dwelling in cities has increased from 17.9% in 1978 to 58.5% in 2017 [7]. This trend has sparked an increasing demand for greater attention to be directed towards the challenges faced by rural China [8]. For example, in March 2014, the government issued the National New-type Urbanization Plan (2014-2020), which advocates that urbanization should be changed from being land-centered to people-oriented [9,10]." (See Lines 64-71)

Response to reviewer 2:

Point 1: Title and abstract – The title is very descriptive but not overly wordy. It needs to go wider for a more specific representation of the study's aims.

Response 1: We appreciate your feedback and suggestions about the title. As required, we have made the following revision.

" Exploring the Impact of College Graduates' Place Attachment on Entrepreneurial Intention upon Returning to Hometowns: A Study Based on the Theory of Planned Behavior" (See Lines 4-6)

Point 2: Please make your abstract attractive to readers (simple sentences without any repetition) and include 2-3 sentences ready to be cited exactly as they are. In 1 paragraph, your abstract should tell the readers why the study is important (maximum 25% of the text), what you did, i.e. your methodology (maximum 25% of the text), and what you found, i.e. main research results and their major implications (50% of the text). This is very important to promote your work because of the growing trend that authors use Google search to find and cite papers based on the abstract (instead of reading the full paper).

Response 2: We are thankful for the reviewer’s comments and suggestions on the abstract. As required, we have the following revisions.

“The issue of the continuing decline of rural areas caused by urbanization has become a global concern. Encouraging college graduates to return to their hometowns to start businesses is an important initiative for countries to achieve sustainable rural development. Drawing from the Theory of Planned Behavior (TPB), this study introduces two additional variables: place attachment and entrepreneurial self-efficacy. Through a series of three model refinements, a comprehensive theoretical framework has been formulated to elucidate Chinese college graduates’ hometown-based entrepreneurial intention and behavior. The samples for this study were 1151 college graduates selected from diverse universities across China. This study aims to explore the influence of their hometown-based entrepreneurial intention using Structural Equation Modelling. This analytical approach illuminates how variables such as college graduates’ place attachment, entrepreneurial self-efficacy, subjective norm for hometown-based entrepreneurship, and attitude towards hometown-based entrepreneurship affected their hometown-based entrepreneurial intention. The research findings reveal the following insights; (1) The overall levels of college graduates’ place attachment and hometown-based entrepreneurial intention were not high. (2) College graduates’ place attachment, entrepreneurial self-efficacy, subjective norm for hometown-based entrepreneurship, and attitude towards hometown-based entrepreneurship, had a positive impact on their hometown-based entrepreneurial intention. (3) College graduates’ place attachment and subjective norm for hometown-based entrepreneurship had a significant impact on their hometown-based entrepreneurial intention through the mediating variable of entrepreneurial self-efficacy. This study then makes policy recommendations from the three aspects including economic inputs, cultural development, and educational practice.” (See Lines 28-49)

Point 3: Regarding the Introduction, the first two paragraphs need to be enhanced with more arguments exemplifying the importance of entrepreneurship and this research. There is a lot of published work in 2021 and 2022, such as Barba et al. (2022) https://doi.org/10.1016/j.iedeen.2021.100184 or Calderon-Milan et al. (2020), https://doi.org/10.3390/su12031079, to strengthen the narrative on this.

Response 3: We are grateful for the reviewer’s comments and suggestions about this. As required, we have made the following revision.

“…… Entrepreneurial activity is recognized as a vital endeavor that merits encouragement due to its significant contributions to the economic and social development of a given area. It serves as a means of creating employment opportunities and plays an essential role in addressing this issue [3]. For example, China has been steadfast in its pursuit of the 'rural revitalization' strategy since 2017. The primary objective of this strategy is to rectify the growing imbalance between rural and urban areas, ultimately striving for more sustainable development. A pivotal approach is to attract both the population and other factors towards rural regions [4]. Consequently, promoting the return of college graduates to their hometowns to initiate businesses emerges as a crucial initiative for nations seeking sustainable rural development.” (See Lines 54-63)

"The promotion of entrepreneurial behavior stands as a significant matter of shared concern among governments and scholars across the globe [11]. The academic community has conducted a substantial number of studies related to the interpretation of entrepreneurial behavior. Among these, the Theory of Planned Behavior (TPB) holds a preeminent position [12, 13]. There is a consensus in the literature regarding the validity of intentional models for predicting the entrepreneurial drive, as intentions are understood as antecedents of actual behavior [14]. In order to enhance the college graduates' entrepreneurial behaviors after returning to hometowns, the paramount objective revolves around identifying the variables that exert influence—directly or indirectly—over the entrepreneurial intentions of these graduates." (See Lines 72-80)

Point 4: In the introduction, it is very worthwhile to introduce readers in more detail to the social and cultural context in which the university student finds himself.

Response 4: Thanks for your suggestion. Indeed, the description of the students' social and cultural context were neglected in the original manuscript. We have now made the following revisions.

"The questionnaires for this study were distributed to 1363 college graduates, who were selected using the convenient sampling method from nine different types of colleges and universities. These nine universities are situated in the western, central, and eastern regions of China, which further enhances the credibility of the research results. These students encompass graduates from different academic backgrounds, such as junior college, undergraduate, and graduate studies. Given that employment choices hold greater significance for these graduates, they are likely to contemplate whether to return to their hometowns for entrepreneurial endeavors." (See Lines 93-100)

Point 5: Where are the contributions of this paper? A clear and focused reference to the innovation and uniqueness of the current study is needed in light of the large number of studies already done in this field.

Response 5: We are grateful for your commens and suggestions about this. As required, the revision are as follows.

" The contribution of this study is as follows. Returning to Hometown for (RHF) entrepreneurship is the focus of our study. The ongoing decline of rural areas caused by urbanization was the source of this problem. While numerous domestic and foreign investigations have concentrated on college students' entrepreneurial intentions and behaviors, there have been fewer studies centered around 'Returning to Hometown For' (RHF) entrepreneurship. Furthermore, the influencing mechanism of subjective norm, attitude and entrepreneurial self-efficacy to intention for RHF entrepreneurship has been verified by a considerable number of studies. In addition to considering that aspect, this study incorporates the Theory of Man-Land Relationship into TPB. From the perspective of environmental psychology, this study has modified the TPB model by incorporating two factors including place attachment and self-efficacy, and employed the TPB model in interpreting college graduates’ hometown-based entrepreneurial intention." (See Lines 101-112)

Point 6: I require that you add text for each one of your hypotheses and justify each one of them by separate and in a better manner you do now. Please try you separate the particular text that corresponds with one or another hypothesis.

Response 6: Thanks for your advices about this part. We have revised it in alignment with your suggestion, which made the hypotheses clearer and more readable.

Point 7: A very important intervening variable is missing- perceived behavioral control. Please explain.

Response 7: We are grateful for the reviewer's suggestion about this. "Perceived behavioral control" is a concept in TPB. Some studies have compared the components of self-efficacy and perceived behacorial control, and noted that self-efficacy may be a more suitable predictor of intention and behavior. To clarify this point, we have included an explanation in the text of hypothesis 3 in Lines 159-170.

Point 8: In the methodology, the authors did not properly present the process of using "face-to-face interviews" as a tool for gathering the information - how was it done, who were the interviewers, what was their training process, how was the interviewer's reliability tested, what were the guiding questions? That is, much more details are required than have been presented.

Response 8: We appreciate the reviewer's suggestion. The following revisions have been made to the manuscript.

" The questionnaires for this study were distributed to 1363 college graduates during October 2020 to December 2020. They were selected by employing the convenient sampling method from nine different types of colleges and universities. These nine universities are situated in the western, central, and eastern regions of China, which further enhances the credibility of the research results. These students encompass graduates from different academic backgrounds, such as junior college, undergraduate, and graduate studies. Given that employment choices hold greater significance for these graduates, they are likely to contemplate whether to return to their hometowns for entrepreneurial endeavors. The questionnaires are adapted from the mature scale. Finally, 1151 valid questionnaires were obtained, with a valid rate of 84.45%." (See Lines 244-253)

Point 9: An article must be self-contained, i.e. it must contain all the information necessary for its comprehension. How the sample has been chosen and to demonstrate that it is representative of the society being studied. I would also recommend adding a table with accurate information about the technical specifications of the study and another table with the comparative demographics between the population and sample.

Response 9: We are grateful for your comments and suggestions about this. Regarding the first point about the description of sampling, we have made the following revision.

" The questionnaires for this study were distributed to 1363 college graduates during October 2020 to December 2020. They were selected by employing the convenient sampling method from nine different types of colleges and universities. These nine universities are situated in the western, central, and eastern regions of China, which further enhances the credibility of the research results. These students encompass graduates from different academic backgrounds, such as junior college, undergraduate, and graduate studies. Given that employment choices hold greater significance for these graduates, they are likely to contemplate whether to return to their hometowns for entrepreneurial endeavors. The questionnaires are adapted from the mature scale. Finally, 1151 valid questionnaires were obtained, with a valid rate of 84.45%. " (See Lines 244-253)

As for the second point about adding a table with accurate information about the technical specifications of the study, we have added fit criteria in Table 1, as follows.

Table 1. Measurement Model Fitting Results.

Latent variables X2/df CFI GFI IFI NFI RMSEA

Fit criteria <4.000 >0.900 >0.900 >0.900 >0.900 <0.08

PA 2.851 0.921 0.894 0.937 0.960 0.05

ESE 2.970 0.915 0.906 0.933 0.901 0.09

SHE 2.762 0.915 0.925 0.913 0.919 0.074

AHE 2.684 0.902 0.914 0.937 0.925 0.055

HEI 2.344 0.912 0.935 0.917 0.915 0.058

Concerning the last point regarding the addition of another table, we apologize for the difficulty in comparing the population and the sample. This is beacause that accurate and comprehensive population information cannot be obtained due to the limited access to information.

Point 10: Please add the validity and reliability of the measurement instruments. It is not enough to report only the Cronbach's alpha coefficient.

Response 10: Thanks for your comments and suggestions for this. In fact, apart from the Cronbach's alpha coefficient, we have reported the validity of the measurement intruments. However, we recognized the need to readjust the structure of the article. As a result, the section pertaining to the measurement instrument validity has now been appropriately repositioned within the Method section.

Point 11: In addition, I would recommend adding a table, that shows a list of items used to measure each of the hypotheses.

Response 11: We are grateful for your comments and suggestions about this. Based on your feedback, we have inceased the representative items for each variable included in the hypotheses. The details are as follows.

" For example, 'after leaving, my hometown is still an important place in my life', 'a sense of intimacy cannot be found in the other place other than my hometown', 'I am familiar with most parts of my hometown', and 'I rely on my hometown'." (See Lines 276-278)

" For example, 'returning to hometown for entrepreneurship is my career goal', and 'if the job is not deal, I will return to hometown for entrepreneurship'." (See Lines 293-294)

"

Point 12: In section 3 it would be better if the control variables used were explained in developing the hypothesis. Where are the results for the control variables used?

Response 12: Thanks for your comments and suggestions. As required, we have made the following revisions. The term 'Control variables' has been deleted from the 'measures' section. Although control variables are difficult to explain in the hypotheses, we have further discussed them in the results section. We greatly appreciate your input, and in light of this, we are contemplating incorporating an explanation of the control variables when formulating hypotheses in our future studies.

" This study found that demographic variables such as age and education background can influence entrepreneurial attitude and self-efficacy. This finding aligns with a previous study that indicated college graduates with higher ages and more extensive educational backgrounds are more adaptable [56]. The author suggests that as college graduates' age and educational background increase, so does their personal capability, leading to a more open career concept. Fueled by social and family responsibilities, individuals may increasingly seek self-realization through entrepreneurial behaviors and plans. Additionally, our research identifies gender as a factor influencing entrepreneurial attitudes, consistent with prior studies [57]. One plausible interpretation is that in China, individuals of different genders often assume distinct social responsibilities. " (See Lines 498-507)

Point 13: The use of SEM should be able to test the models that have been developed. Unfortunately, how the initial model and the final model after testing is not fully shown. It would be better if this could be done so that the placement of variables or the constructs that build them could be analyzed. Please show the variables with their dimensions.

Response 13: We are grateful for your comments and suggestions for this. Upon conducting testing, we have observed differences between the final model and the initial model. Specifically, in section 4, we discovered that H5 has not been verified. This indicates that the attitude towards RHF entrepreneurship had no statistically significant impact on entrepreneurial self-efficacy. As a result, we have depicted the path from attitude to entrepreneurial self-efficacy using a dotted line. As required, we have included the variables along with their respective dimensions. The aforementioned revisions are outlined below.

Point 14: The conclusion and discussion section lack significant theoretical reference to the findings. The research should relate to/correspond with theories of entrepreneurship, to the Theory of Planned Behaviour.

Response 14: We greatly appreciate your comments and suggestions regarding the conclusion and discussion sections. As required, we have made comprehensive adjustments to this portion. For instance, we have enhanced the section's readability, enriched the theoretical references related to the findings, and expanded the discussion on two theories. You can find detailed information in lines 448 to 563.

---

## [Decision Letter · Decision Letter 1]

22 Nov 2023

PONE-D-23-12351R1Exploring the Impact of College Graduates' Place Attachment on Entrepreneurial Intention upon Returning to Hometowns: A Study Based on the Theory of Planned Behavior.PLOS ONE

Dear Dr. Xu,

Thank you for submitting your manuscript to PLOS ONE. After careful consideration, we feel that it has merit but does not fully meet PLOS ONE’s publication criteria as it currently stands. Therefore, we invite you to submit a revised version of the manuscript that addresses the points raised during the review process.

We look forward to receiving your revised manuscript.

Kind regards,

Tien-Chi Huang

Academic Editor

PLOS ONE

Journal Requirements:

Reviewers' comments:

Reviewer's Responses to Questions

**Comments to the Author**

1. If the authors have adequately addressed your comments raised in a previous round of review and you feel that this manuscript is now acceptable for publication, you may indicate that here to bypass the “Comments to the Author” section, enter your conflict of interest statement in the “Confidential to Editor” section, and submit your "Accept" recommendation.

Reviewer #2: All comments have been addressed

Reviewer #3: All comments have been addressed

Reviewer #4: (No Response)

2. Is the manuscript technically sound, and do the data support the conclusions?

Reviewer #2: Yes

Reviewer #3: Yes

Reviewer #4: Yes

3. Has the statistical analysis been performed appropriately and rigorously? 

Reviewer #2: Yes

Reviewer #3: Yes

Reviewer #4: Yes

4. Have the authors made all data underlying the findings in their manuscript fully available?

Reviewer #2: No

Reviewer #3: Yes

Reviewer #4: Yes

5. Is the manuscript presented in an intelligible fashion and written in standard English?

Reviewer #2: Yes

Reviewer #3: Yes

Reviewer #4: No

6. Review Comments to the Author

Reviewer #2: (No Response)

Reviewer #3: Dear authors,

I hope this letter finds you well. I am writing to convey my assessment of the manuscript titled "Exploring the Impact of College Graduates' Place Attachment on Entrepreneurial Intention upon Returning to Hometowns: A Study Based on the Theory of Planned Behavior," which was submitted for consideration to PLOS ONE.

Upon thorough review of the manuscript and all of the comments provided by the peer reviewers, I have determined that the authors have done a commendable job in addressing and incorporating the feedback. The revisions made in response to the reviewers' comments have significantly improved the quality and rigor of the research presented in the manuscript.

I appreciate the authors' dedication and effort in revising their work, and their commitment to addressing all the comments made by the reviewers is evident. The changes made align well with the scope and standards of our journal, and I believe the manuscript is now suitable for publication in its current form.

Considering the comprehensive revisions made and the authors' timely responses to the reviewers' comments, I am pleased to recommend that the manuscript be accepted for publication in PLOS ONE without further revisions. The research and findings presented in the manuscript contribute meaningfully to the field and meet the high standards of this journal.

Reviewer #4: Thank you very much for the opportunity to review this manuscript. I have some suggestions as follow:

Please emphasize and elucidate the significance of your hypotheses in advancing the understanding of the mediating role of self-efficacy. Based on the literature, it is suggested that self-efficacy positively influences outcomes. We hypothesize that self-efficacy will exert an indirect positive effect between X and Y. Kindly provide further elaboration.

Additionally, please offer more insights and details on the demographics of the respondents. Present this information in tabular form.

Separate implications into theoretical and managerial categories.

Proofreading is required for clarity, as some sentences are difficult to understand and contain grammatical issues.

This version aims to enhance clarity and flow, making it easier for readers to grasp the intended meaning.

7. PLOS authors have the option to publish the peer review history of their article (what does this mean?). If published, this will include your full peer review and any attached files.

Reviewer #2: No

Reviewer #3: **Yes: **Mohammad Heydari

Reviewer #4: No

---

## [Author Response · Author response to Decision Letter 1]

23 Jan 2024

Response to Reviewer 4 Comments:

Point 1: Please emphasize and elucidate the significance of your hypotheses in advancing the understanding of the mediating role of self-efficacy. Based on the literature, it is suggested that self-efficacy positively influences outcomes. We hypothesize that self-efficacy will exert an indirect positive effect between X and Y. Kindly provide further elaboration.

Response 1: We sincerely appreciate the reviewer's suggestion. To enhance the clarity and robustness of our study on the mediating role of self-efficacy, we have incorporated essential references and refined the internal coherence of our argument. The revised content is detailed below.

" As established in empirical research, both subjective norm [20] and attitude [21] wield significant influence on the intention for RHF entrepreneurship. Building upon this foundation and our hypotheses, we recognize the pivotal impact that place attachment may exert on the intention for RHF entrepreneurship. Additionally, prior studies has underscored the crucial role of self-efficacy as a predictor of behavior and the strongest determinant of behavioral intention [24,40]. Entrepreneurial self-efficacy, in particular, has been identified as a significant influencer of entrepreneurial intention [41].

Moreover, our review of existing literature corroborates the following points:

Entrepreneurship education can shape entrepreneurial self-efficacy through attitude [41]. Based on this finding, we propose the following hypothesis: 

H5: Self-efficacy has a mediating effect on the relationship between attitude towards RHF entrepreneurship and intention for RHF entrepreneurship.

Secondly, place attachment has a significantly positive influence on entrepreneurial self-efficacy [42]. Accordingly, we posit the following hypothesis:

H6: Self-efficacy has a mediating effect on the relationship between place attachment and intention for RHF entrepreneurship.

Finally, subjective norm has a significantly positive influence on self-efficacy [43]. Building upon this observation, we present the following hypothesis:

H7: Self-efficacy has a mediating effect on the relationship between subject norm for RHF entrepreneurship and intention for RHF entrepreneurship.

Drawing from these analyses, we establish multiple mediating relationships with entrepreneurial self-efficacy serving as the mediating variable within the TPB model. Subsequently, we have refined the model for the third iteration, and the updated theoretical model diagram is depicted in Fig 1." (See Lines 218-241)

Point 2: Additionally, please offer more insights and details on the demographics of the respondents. Present this information in tabular form.

Response 2: We greatly appreciate the reviewer’s comments and suggestions. As required, we have added more details of the demographics of the respondents and presented this information in tabular form. The detailed revisions can be found in the specified lines. (See Lines 259-260)

Table 1. The Demographics of the Respondents.

Variables Level Quantity Ratio

Gender Male 696 60.47%

 Female 455 39.53%

Academic backgrounds Associate degree or below 437 37.97%

 Bachelor’s degree 468 40.66%

 Master’s degree or above 246 21.37%

Professional backgrounds Humanistic and social science 582 45.94%

 Science and Engineering 685 54.06%

Entrepreneurial backgrounds Yes 143 11.29%

 No 1124 88.71%

Point 3: Separate implications into theoretical and managerial categories.

Response 3: We are grateful for the reviewer’s suggestions. We have modified the structure of the Implications section and split it into theoretical and managerial categories, which can be seen from lines 579 to lines 634. 

Point 4: Proofreading is required for clarity, as some sentences are difficult to understand and contain grammatical issues. This version aims to enhance clarity and flow, making it easier for readers to grasp the intended meaning.

Response 4: Thanks for your valuable suggestion. As required, we have re-examined the manuscript and modified the problematic statement. These changes are clearly delineated in the manuscript using the track changes feature. We hope that this version can make it easier for readers to grasp the intended meaning.

---

## [Editor Report · Decision Letter 2]

27 Feb 2024

Exploring the Impact of College Graduates' Place Attachment on Entrepreneurial Intention upon Returning to Hometowns: A Study Based on the Theory of Planned Behavior.

PONE-D-23-12351R2

Dear Dr. Xu,

We’re pleased to inform you that your manuscript has been judged scientifically suitable for publication and will be formally accepted for publication once it meets all outstanding technical requirements.

Kind regards,

Tien-Chi Huang

Academic Editor

PLOS ONE
---

## [Editor Report · Acceptance letter]

20 Mar 2024

PONE-D-23-12351R2 

PLOS ONE

Dear Dr. Xu, 

I'm pleased to inform you that your manuscript has been deemed suitable for publication in PLOS ONE. Congratulations! Your manuscript is now being handed over to our production team.

Kind regards, 

on behalf of

Professor Tien-Chi Huang 

Academic Editor

PLOS ONE